# Identifying Informative Latent Variables Learned by GIN via Mutual Information

## Abstract

How to learn a good representation of data is one of the most important topics of machine learning. Disentanglement of representations, though believed to be the core feature of good representations, has caused a lot of debates and discussions in recent. Sorrenson et al. (2020), using the techniques developed in nonlinear independent component analysis theory, show that general incompressible-flow networks (GIN) can recover the underlying latent variables that generate the data, and thus can provide a compact and disentangled representation. However, in this paper, we point out that the method taken by GIN for informative latent variables selection is not theoretically supported and can be disproved by experiments. We propose to use the mutual information between each learned latent variables and the auxiliary variable to correctly identify informative latent variables. We directly verify the improvement brought by our method in experiments on synthetic data. We further show the advantage of our method on various downstream tasks including classification, outlier detection and adversarial attack defence on both synthetic and real data.

## 1 Introduction

Representation learning is arguably one of the most important area in machine learning. Many researchers believe that being able to extract useful and interpretable features is a crucial advantage of deep networks over other learning models. A data representation can be obtained either via a supervised learning task or an unsupervised learning task. The former one includes the popular ImageNet pretrained backbone in computer vision tasks, while the latter one mainly consists of generative models like variants of VAEs, GANs and flow-based models. Among all the generative models, VAEs and flow-based models can naturally output the representation of data or even their density, which is convenient for the representation learning purpose. Moreover, supervision information of labels can be integrated into generative models to further improve their performance. General Incompressible-flow Networks (GIN, Sorrenson et al. (2020)), the model we considered in this paper, falls into this case.

Disentanglement is a widely discussed concept by many representation learning works. However, to the best of our knowledge, it has not been given a widely accepted definition (Bengio et al., 2013; Higgins et al., 2018). Many disentangled representation learning algorithms focus on recovering the independent latent variables that generate the data (Burgess et al., 2018; Chen et al., 2018b). However, Locatello et al. (2018) show that without more assumptions than independence among latent variables, it is impossible to recover them from the observation of data. This result is equivalent to the non-identifiability of non-linear independent component analysis (ICA) (Comon, 1994). Actually, any assumptions solely on the latent variables' distribution without referring to the observable data is not sufficient for the identifiability (Hyvarinen and Morioka, 2016; Khemakhem et al., 2020).

A set of sufficient conditions is proposed under the framework of nonlinear ICA by Khemakhem et al. (2020). The core condition requires that the data, denoted by $\mathbf{x}$, are generated by latent vectors $\mathbf{z}$ through a generative model $p(\mathbf{x} \mid \mathbf{z})$, and conditioned on an auxiliary variable $u$, the entries of $\mathbf{z}$ are independent and follow some exponential family distributions. This can be expressed by the

following formulas.

$$p_{\mathbf{T},\lambda}(\mathbf{z}|u) = \prod_{i=1}^{n} \frac{Q_i(z_i)}{Z_i(u)} \exp\left[\sum_{j=1}^{k} T_{i,j}(z_i)\lambda_{i,j}(u)\right], \qquad (1)$$

$$\mathbf{x} = f(\mathbf{z}) + \boldsymbol{\epsilon}. \qquad (2)$$

In Eq. 1 and 2, $\mathbf{z} \in \mathbb{R}^n$ and $\mathbf{x} \in \mathbb{R}^d$. Usually people believe that $n \ll d$; that is, the data are distributed near a low dimensional manifold embedded in a very high dimensional space. With both $u$ and $\mathbf{x}$ being observable, the identifiability of $\mathbf{z}$ can be established. Moreover, Khemakhem et al. also show that the latent variables and the generative function $f$ can be learned by optimizing the ELBO using a special VAE, called iVAE (i for identifiable).

However, iVAE assumes that all the latent variables are correlated with $u$, and it requires the correct number of latent variables for iVAE to work, which is usually not known. Sorrenson et al. (2020) consider the generative model with Eq. 2 replaced by

$$\mathbf{x} = f(\mathbf{z}, \boldsymbol{\epsilon}), \qquad (3)$$

where $f$ is invertible, $p_{\mathbf{T},\lambda}(\mathbf{z}|u)$ is a Gaussian density supported on $\mathbb{R}^n$ with free expectations and diagonal covariance, and $p(\boldsymbol{\epsilon})$, supported on $\mathbb{R}^{d-n}$, is free from $u$. Entries of $\mathbf{z}$ are refered to as informative latent variables, while entries of $\boldsymbol{\epsilon}$ are treated as noise. They prove that $\mathbf{z}$ is identifiable up to a permutation and scaling. Based on this identifiability result, Sorrenson et al. propose to learn the informative latent variables $\mathbf{z}$ using a flow-based model, called the General Incompressible-flow Networks (GIN). The output of a GIN model, $\mathbf{w} = g(\mathbf{x})$, is a vector with its dimension equal to $\dim(\mathbf{z}, \boldsymbol{\epsilon})$, since both $f$ and $g$ are invertible. We, hence, need a method to identify the entries of $\mathbf{w}$ that estimate $\mathbf{z}$.

Sorrenson et al. propose a variance based criterion for the informative latent variables selection. Specifically, they select $w_i$'s with large variances as estimates to $\mathbf{z}$ and treat the remaining entries as solely related to $\boldsymbol{\epsilon}$. Sorrenson et al. do not clearly explain the logic behind this criterion. For the following reasons, we think this criterion is not guaranteed to work all the time:

1. Note that a volume-preserving transform cannot preserve the variances of the input data, because volume-preserving constrains the determinant of the Jacobian instead of the size of each singular value.

2. According to the identifiability theorem in Sorrenson et al. (2020), each $z_i$ can only be estimated up to an affine transform, which means that the variances of informative $w_i$'s can be smaller than the variances of non-informative ones.

Nevertheless, they show that the variance based criterion can correctly identify the ground truth informative latent variables in experiments on synthetic data. However, in their experiments, the noise latent variables all have very small variances. When reproducing their experiments, we find that if we increase the variances of $\boldsymbol{\epsilon}$, but which are still less than the variances of $\mathbf{z}$), this criterion will fail.

On the other hand, the identifiability result in Sorrenson et al. (2020) implies that the entries of $\mathbf{w}$ can be categorized into two classes, entries in one class are linearly related to entries of $\mathbf{z}$, while entries in the other class have distributions that are irrelevant to $u$. This observation inspires us to scrutinise the notion of informative latent variables under the generative framework described by Eq. 3. *The key difference between informative and non-informative latent variables is not their variances, but whether they are correlated with the auxiliary variable $u$.*

Following this new perspective, it is natural to choose the mutual information between $w_i$'s and $u$ as the criterion for identifying informative latent variables, which we call it the mutual information (MI) criterion. In this paper, we compare the MI criterion with the previous variance based criterion (VAR) using abundant experiments on various tasks, including disentanglement quality on synthetic data, classification performance on real data, outlier detection capability, and robustness under adversarial attacks. In all these cases, the MI criterion shows superior performance over the VAR criterion.

The main content of the paper is organised as follows. In Section 2, we briefly review the identifiability result of Sorrenson et al. (2020), and show why the mutual information can work for

informative latent variables selection. In Section 3, we compare the performance of the MI criterion with that of the VAR criterion by experiments on synthetic data generated following the theoretical assumptions. In Section 4, we demonstrate the impacts of different informative variables selection criteria on classification, outlier detection, and defence against adversarial attacks by experiments on EMNIST (Cohen et al., 2017). In Section 5, we discuss some other interesting discoveries related to the representation learning under the framework of nonlinear ICA. And finally, we conclude our results and summarise some future working directions in the last section.

## 2 THEORY OF NONLINEAR ICA AND MUTUAL INFORMATION CRITERION

Since our work is an improvement based on GIN, we first review the main result of Sorrenson et al. (2020) with a detailed elaboration on the relation between the learned representation $\mathbf{w}$ and the auxiliary variable $u$, and then make use of this property to derive our mutual information criterion. Due to the length restriction of the paper, we refer interested readers to Appendix A for formal assumptions and statements.

With the data generation model defined by Eq 1 and 3 The proof of the identifiability theorem in Sorrenson et al. (2020) mainly consists of the following three steps:

1. Start with $p_\theta(\mathbf{x} \mid u) = p_\phi(\mathbf{x} \mid u)$, where the conditional marginal data distributions are determined by Eq. 1 and Eq. 3 with two possibly different transforms, latent variables and parameters; $\theta = (f, \mathbf{T}(\mathbf{z}, \boldsymbol{\epsilon}), \lambda)$ in the generation process and $\phi = (g, \mathbf{T}(\mathbf{w}), \lambda')$ in the estimation process.

2. Then eliminate terms in Eq. 1 that are irrelevant to the auxiliary variable $u$.

3. And finally obtain the relation between the statistics $\mathbf{T}(\mathbf{z})$ and $\mathbf{T}(\mathbf{w})$,

$$\mathbf{T}(\mathbf{z}) = L^{-1}L'\mathbf{T}(\mathbf{w}) + \mathbf{c}\,,$$

where

$$L = \begin{bmatrix} \lambda_{1,1}(u_1) - \lambda_{1,1}(u_0) & \lambda_{1,2}(u_1) - \lambda_{1,2}(u_0) & \cdots & \lambda_{n,k}(u_1) - \lambda_{n,k}(u_0) \\ \vdots & \vdots & \ddots & \vdots \\ \lambda_{1,1}(u_{nk}) - \lambda_{1,1}(u_0) & \lambda_{1,2}(u_{nk}) - \lambda_{1,2}(u_0) & \cdots & \lambda_{n,k}(u_{nk}) - \lambda_{n,k}(u_0) \end{bmatrix}$$

$$L' = \begin{bmatrix} \lambda'_{1,1}(u_1) - \lambda'_{1,1}(u_0) & \lambda'_{1,2}(u_1) - \lambda'_{1,2}(u_0) & \cdots & \lambda'_{d,k}(u_1) - \lambda'_{d,k}(u_0) \\ \vdots & \vdots & \ddots & \vdots \\ \lambda'_{1,1}(u_{nk}) - \lambda'_{1,1}(u_0) & \lambda'_{1,2}(u_{nk}) - \lambda'_{1,2}(u_0) & \cdots & \lambda'_{d,k}(u_{nk}) - \lambda'_{d,k}(u_0) \end{bmatrix}$$

and entries of $\mathbf{c}$ are functions of $u$.

Since the latent variables conditioned on $u$ are assumed to be distributed as Gaussian, $\mathbf{T}(\mathbf{z}) = (\mathbf{z}, \mathbf{z}^2)^\top$. Their main theorem shows that after appropriate rearrangement of entries of $\mathbf{z}$,

$$L^{-1}L' = \begin{bmatrix} D & \overbrace{\mathbf{0}}^{n \times (d-n)} & \mathbf{0} & \mathbf{0} \\ \mathbf{0} & \mathbf{0} & D^2 & \mathbf{0} \end{bmatrix}, \tag{4}$$

where $D$ is an $n \times n$ diagonal matrix with non-zero entries. This means that after rearrangement, the first $n$ entries of $\mathbf{w}$ are just scaling of $\mathbf{z}$. For the rest $d - n$ entries, denoted by $\mathbf{w}_{n+1:d}$, their connection with $\mathbf{z}$ cannot be justified by Eq. 4, since the corresponding coefficients are zeros. Their relation with $u$, however, is clear. Since entries in $L^{-1}L'$ corresponding to $\mathbf{w}_{n+1:d}$ are all zero, we know that $\lambda'_{ij}(u_l)$ must equal to $\lambda'_{ij}(u_0)$ for $i \in [n+1, d]$, $j = 1, 2$ and $l \in [1, nk]$. This means that the distribution of $w_i$ for $i \in [n+1, d]$ remains unchanged no matter what is the value of $u$.

Based on their result, we know that an informative $w_i$ equals to some $z_j$ after an affine transform and thus has non-zero mutual information with $u$, while non-informative $w_i$ has zero mutual information with $u$. Therefore, the mutual information between each $w_i$ and $u$ can be used as a reliable indicator for justifying whether $w_i$ is an informative latent variable. On the other hand, as we explained in the introduction, the composite mapping $g^{-1} \circ f$ does not necessarily preserve the scale of $\mathbf{z}$ and $\boldsymbol{\epsilon}$. Even if $\boldsymbol{\epsilon}$ have much smaller variances compared to $\mathbf{z}$, it is not necessarily true that the variances of $\{w_i\}_{i=n+1}^d$ are less than those of $\{w_i\}_{i=1}^n$. We demonstrate this by an experiment in the next

section. Therefore using the size of variances of $\{w_i\}_{i=1}^d$ for informative latent variables selection can result in serious problems.

We summarise the algorithm for the MI criterion in Alg. 1. The mutual information between $w_i$ and $u$ is computed based on the samples by directly calling the function provided in the python package `sklearn`. To choose the threshold for mutual information decay, one can employ existing techniques for truncation, i.e., maximum curvature, sharp drop, etc. The computational cost of mutual information is usually huge for high dimensional random vectors, but for that between a continuous scalar random variable and a discrete random variable, the computational cost is negligible compared to network training time.

---

**Algorithm 1:** Informative Latent Variables Selection (MI)

    **input** : Learned latent variables $\{w_i\}_{i=1}^d$, auxiliary variable $u$ and initialise $T = \emptyset$
    **output:** Set of indices of informative latent variables $T$

**1** **for** $i \leftarrow 1$ **to** $n$ **do**
**2**     Evaluate $I(w_i; u)$;
**3** Choose thresholds $\eta > 0$ according to the decay of $\{I(w_i; u)\}_{i=1}^n$;
**4** **for** $i \leftarrow 1$ **to** $n$ **do**
**5**     **if** $I(w_i; u) > \eta$ **then** Add $i$ into $T$;

---

## 3 PERFORMANCE ON SYNTHETIC DATA

In this section, we demonstrate the performance of the MI criterion on synthetic data, compared to the VAR criterion. The synthetic data are constructed similarly with the GIN paper. The data space is $\mathbb{R}^{10}$ and so is the latent space. Within the 10 latent variables, there are only 2 informative ones. These two latent variables $z_1, z_2$ are distributed as a Gaussian mixture with 5 components, each of which, labelled by the auxiliary variable $u$, has a different expectation and a diagonal covariance matrix. The extra 8 latent variables $\epsilon$ are treated as noise, which follows independent Gaussian distributions. An invertible function $f : \mathbb{R}^{10} \rightarrow \mathbb{R}^{10}$ maps each latent vector sampled according to the distribution described above to a data point $\mathbf{x}$. In our experiment, the statistics of the Gaussian mixture are randomly generated (see Appendix D), the noise variables are distributed as the standard Gaussian, and the invertible function is simulated by a flow model with random parameters. Note that in our data generating process, the variance of the noise is still less than that of informative variables, but not as small as in Sorrenson et al. (2020). Figure 1 shows 4 examples of the distributions of the 10 dimensional latent vectors projected to different choices of variable pairs. When they are projected to the correct informative dimensions, the illustration shows a typical Gaussian mixture shape, as shown in the left subfigure. The other three subfigures show the distribution of latent variables when only one or none of the projection directions is informative. Figure 2 shows the distribution of data $\mathbf{x}$ projected to different choices of dimensions.

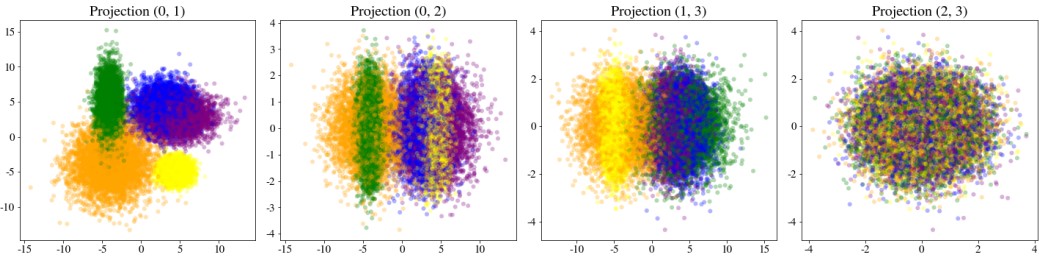

Figure 1: Four 2D-projections of the distribution of latent variables $(\mathbf{z}, \epsilon)$. Projection $(i, j)$ means the 2D-projection to the $i$-th and $j$-th dimension. $(0, 1)$ are the informative latent variables $\mathbf{z}$. Points sampled from 5 different Gaussian components in the Gaussian mixture are dyed by different colours.

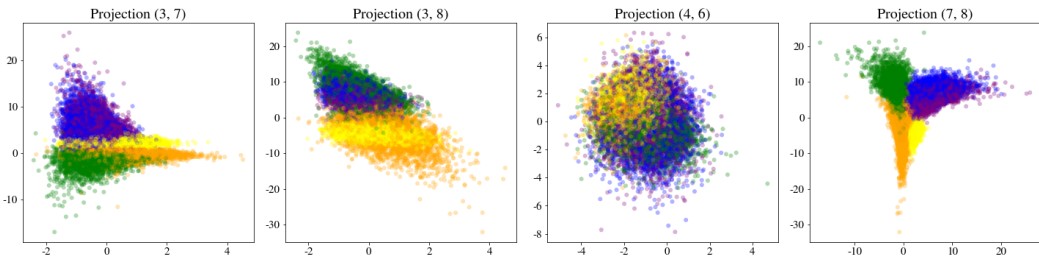

Figure 2: Four 2D-projections of the data $\mathbf{x} = f(\mathbf{z}, \boldsymbol{\epsilon})$.

To learn the representation from data, we train a GIN model by minimising the following objective:

$$\mathcal{L}(\theta, \boldsymbol{\mu}, \boldsymbol{\sigma}) = \mathbb{E}_{(\mathbf{x}, u) \in \mathcal{D}} \left[ \frac{1}{n} \sum_{i=1}^{n} \left( \frac{\left( g^{-1}(\mathbf{x}; \theta) - \mu_i(u) \right)^2}{2\sigma_i^2(u)} + \log(\sigma_i(u)) \right) \right]. \quad (5)$$

Note that here both the weights of the flow $\theta$ and the statistics of the Gaussian mixture $(\boldsymbol{\mu}, \boldsymbol{\sigma})$ are tuned during training. Another form of objective suggested for the GIN model in the original paper is to replace trainable $(\boldsymbol{\mu}, \boldsymbol{\sigma})$ by batch statistics. Consistent with the report of the original paper, we find that setting $(\boldsymbol{\mu}, \boldsymbol{\sigma})$ to be trainable shows similar performance with equating them to batch statistics. However, trainable $(\boldsymbol{\mu}, \boldsymbol{\sigma})$ match the theoretical framework better, avoid large variation caused by small batch size, and save the computational cost of evaluating the statistics for density estimation during downstream tasks. Therefore, in the main content of the paper, we will always consider the objective given by Eq 5. For the experiment results of the alternative way, see Appendix C. Other specific choices of parameters for the training include: batch size at 240 and initial learning rate at $10^{-2}$ with multiplicative decay rate 0.1.

After training, we compute both the variances of the features in the representation $\mathbf{w}$ and their mutual information with the auxiliary variable $u$, and plot them in descending order; see Figure 3. If we select the two latent variables with the largest variances and plot them, we can see that their distribution is different from the informative latent variables as shown in Figure 1. According to the MI criterion, we select the two variables with the largest mutual information with $u$. They almost perfectly reproduce the distribution of the 2 informative latent variables. If we do not know the correct number of informative variables, we can estimate by truncating the number of features in the representation according to the magnitude of mutual information values. In Figure 3, we can see that the sharp drop occurs exactly between the second and the third variables from left. This experiment shows that the GIN model can successfully learn the informative latent variables as indicated by the theory, but we need the MI criterion to correctly identify them.

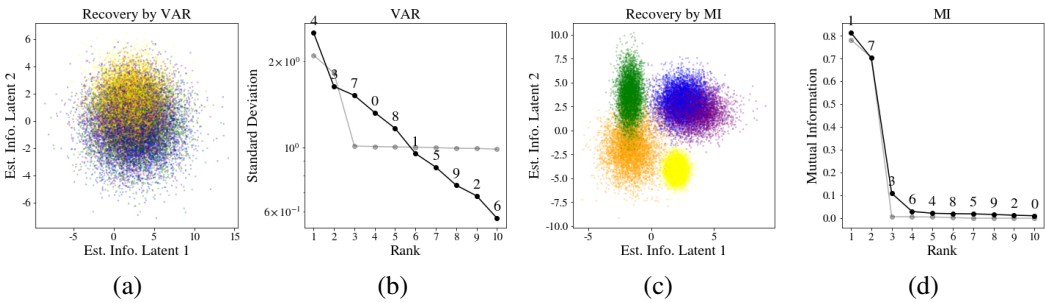

Figure 3: Performance of the VAR criterion and the MI criterion on the informative latent variables identification on synthetic data. a) The distribution of the 2 features with the largest variances; b) The standard deviations of the 10 features (black) of $\mathbf{w}$ and the 10 true latent variables (grey) in descending order. The numbers are the indices of features for convenience of comparison. c) The distribution of the 2 features with the largest mutual information with $u$. d) The mutual information of the 10 features (black) of $\mathbf{w}$ and the 10 true latent variables (grey) in descending order.

We further demonstrate the importance of identifying the correct informative latent variables for the classification and outlier detection tasks. Note that in our experiments $u$ is the label in the classification task. For the classification task, we evaluate the posterior probability $p_S(u \mid \mathbf{x})$ for each test example using the features of the representation selected into the set $S$ according to either our MI criterion or the VAR criterion. Specifically, the posterior distribution of label $u$ is defined as

$$p_S(u|\mathbf{x}) = p(u|\mathbf{w}_S) \propto p(\mathbf{w}_S|u)p(u), \tag{6}$$

where $\mathbf{w}_S$ is the vector of selected features, $p(\mathbf{w}_S|u)$ is derived from $p(\mathbf{w}|u)$ that learned by the GIN model and $p(u)$ is the distribution of labels (uniform distribution of 5 classes). In Figure 4, we can see that using the two latent variables identified by our MI criterion, the classification tasks can be solved almost perfectly, while the original VAR criterion is not competent at all.

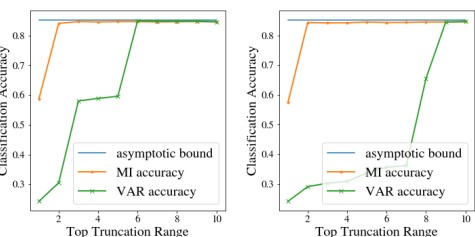 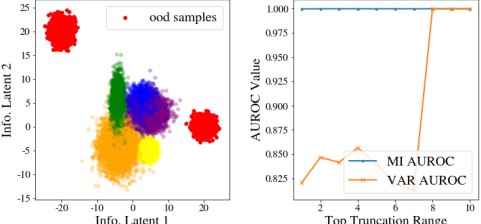

Figure 4: Performance on synthetic data classification: MI criterion vs VAR criterion. The x-axis represents the number of features selected for the $p_S(y \mid \mathbf{x})$ calculation according to the MI and VAR criterion respectively; the y-axis for the accuracy. The asymptotic bound is the theoretical upper bound for the classification accuracy. 2 subfigures show 2 different trials.

Figure 5: Left: The illustration of out-of-distribution samples (in red, from a mixture of two Gaussians) samples from ground truth Gaussian mixture in the latent space projected to the informative dimensions. Right: The AUROC values for different number of features selected by the MI criterion and the VAR criterion respectively.

For the outlier (or out-of-distribution sample) detection task, we sample outliers in the latent space from two Gaussians far apart from the high-density area of normal latent samples, as shown in Figure 5, and then send them through the generator to obtain the data outliers. The test set is constructed by mixing nearly equal number of normal samples and outlier samples. We choose the density evaluated on the chosen $w_i$'s as the criterion for detection. The values of area under receiver operating characteristic (AUROC) in Figure 5 reflect the effectiveness of the criterion in outlier detection (Hendrycks and Gimpel, 2016). The MI criterion again shows significant advantages over the VAR criterion.

In both the classification and the outlier detection tasks, it seems that involving all the latent variables can still work well, though we may pay a huge computational price when the dimension of data is high. The situation is different in practice. When too many noise variables are involved in the density evaluation, the signal provided by the informative latent variables may not be able to outstand. This becomes clear in our experiments on EMNIST in the next section.

## 4 PERFORMANCE ON EMNIST

In this section, we investigate the performance of the MI criterion on the EMNIST dataset. The models are trained on the EMNIST-digits dataset with the same network architecture adopted by Sorrenson et al. (2020) and the objective function shown in Eq 5. For this experiment, the data $\mathbf{x} \in \mathbb{R}^{28 \times 28}$ are the digits images, the auxiliary variable $u$ is the label of digits and the learned feature vector $\mathbf{w}$ is 784 dimensional.

The real dataset by no means satisfies the exact assumptions required by the identifiability theorem. And hence experiments on real datasets can help us understand how much of reality these assumptions capture. Since the underlying latent variables are not available for real datasets, we cannot directly verify the performance of the representation learning and the informative latent variable selection algorithms by visualising their projected distribution. A common method is to visualise the

sample generation quality by traversing the latent space to demonstrate the performance. But this is still very qualitative and arbitrary to subjective interpretation. Instead of these visualisation methods, we choose three downstream tasks: classification, outlier detection and adversarial defence, to measure the performance of the algorithm.

The classification tasks are solved in a similar way as in last section by computing the posterior probability of labels conditioned on the test samples using features selected by each criterion. The result is shown in the left subfigure of Figure 6. With 30 latent variables identified by the MI criterion, the classification accuracy can reach $94.15\%$, while using VAR criterion cannot gives us satisfactory accuracy. Different from the phenomenon we observed on synthetic data, the accuracy degrades when we select too many features that have small mutual information with the auxiliary variable. This indicates that the noise in the representation extracted from data may hurt the performance of downstream tasks, and thus accurately selecting the informative variables is crucial. From the right subfigure, we can see that the mutual information curve has a sharp decay point, which corresponds to the number of latent variables that gives the best classification performance.

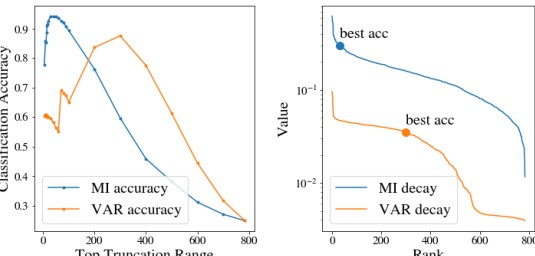 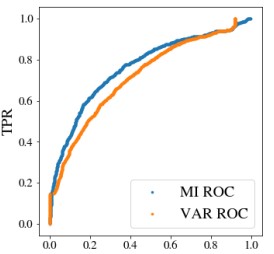

Figure 6: Left: the classification accuracy on EMNIST-digits of the MI criterion and the VAR criterion with different possible top truncation within the full range. Right: the decay curves of mutual information and standard deviation of the estimated latent. The truncation for best accuracy: 30 for MI; 300 for VAR.

Figure 7: The ROC curves using the density value (on the learned Gaussian mixture) as threshold for outlier detection task, MI30 versus VAR300.

In contrast, recall that on synthetic data, involving noise variables does not hurt the classification accuracy. Our conjecture is that the GIN model trained on the synthetic data almost perfectly disentangled the informative and noise latent variables, which can be seen from the sharp decay of mutual information in Figure 3. Since the noise features in the representation contain little information of $u$, they cannot impact the calculation of posterior probability $p(u \mid \mathbf{x})$. Therefore, in this case, involving noise variables does not hurt the performance. However, for the model trained on real data, the features it learned are not all informative at the same level. We can see this from the right subfigure of Figure 6, where many latent variables have mutual information with $u$ around the level of 0.1. These variables may not be able to provide more information about the labels than the first 30 variables, but they may affect the predicted labels. In particular, if the marginal distributions of these variables have small variances; that is, all components are clustered close with each other, then a small value shift on these variables will cause a huge change of density, which can dominate the overall density and change the predicted label. This also explains why the variance criterion works better when more than 200 variables are included: The variables with relatively large variances are not likely to strongly affect the overall density. This conjecture can be further investigated in the future. Nevertheless, it is worth noting that as the low mutual information entries are very unlikely to encode additional information than those with high mutual information, and thus we should not include them for downstream tasks in principle.

To investigate the outlier detection capability, we use the model trained on EMNIST-digits to distinguish EMNIST-letters from EMNIST-digits samples. Using the density as the detection threshold, we plot the ROC curves for the 30 variables selected by the MI criterion, referred to as MI30, and the 300 variables selected by the VAR criterion, referred to as VAR300. The number of latent variables are chosen at the best classification accuracy level. Figure 7 shows that MI30 dominates the VAR300. The AUROC value of the MI30 is 0.763, while the value of VAR300 is 0.591.

Adversarial examples are an extreme case of outliers. These examples deviate from normal samples so little that human vision cannot identify the difference, but they can fool the classification model and change its prediction. Li et al. (2019) shows that conditional variational autoencoders are more robust than discriminative models against adversarial attacks when used together with the Bayes rule for the classification purpose and carefully designed defence mechanism. Here we are not trying to show if the GIN is a better model for adversarial robustness than other probabilistic models. Instead, we want to show that using the mutual information criterion can help GIN models achieve better adversarial robustness.

In the experiment, we attack the model with the widely-used fast gradient sign method (FGSM) (Goodfellow et al., 2015). By attempting adversarial attack at different levels, we find out that without applying defence, the classifier based on $p_S(u \mid \mathbf{x})$ is not robust no matter for MI30 or VAR300. For the model using MI30, the optimal attacking strength is $\eta = 0.02$. As shown in Table 1, at that level, the prediction of the model can be changed by nearly $99\%$ adversarial examples. At the same level, the model using VAR300 can only be successfully fooled by $56\%$ examples. However, we cannot simply conclude that VAR300 should be preferred than MI30 because its clean accuracy is much lower and $\eta = 0.02$ is not the optimal attacking strength for VAR300. To defend the adversarial attack, we define the following rejection rule: If $p_S(\mathbf{w}(\mathbf{x})|u) < p_S(\boldsymbol{\mu}_y - k\boldsymbol{\sigma}_u|u)$ (here $k = 1$ by our choice) for all label $u$, the criterion reports $\mathbf{x}$ as an adversarial example and refuses to classify it. Note that for MI30 and VAR300, their subset $S$ of features are different. At the level $\eta = 0.02$, this defence strategy on MI30 can recognise about $94\%$ attacks and thus only $5.8\%$ of all attacks can succeed. If the attacker wants to bypass the defence, it has to choose a much weaker attacking strength and the overall success rate is still quite low. This is the case for MI30 when $\eta = 0.001$ as shown in Table 1 . However, if we use VAR300, such a defence strategy has almost no effect at level $\eta = 0.001$ and very limited effect at level $\eta = 0.02$. An interesting observation on the no attack case ($\eta = 0$) is that the error rate after applying the defence module is slightly lower than the error rate without defence. Though it does not mean that the overall classification accuracy is improved because around $1 - 0.916\% = 0.084\%$ test samples are screened out and do not get their predicted labels, it does imply that these test samples are overall harder for the model, which is not confident in its predictions on them. For VAR300, we did not observe such an effect. All these results indicate that for defending adversarial attacks, features selected by the MI criterion are more reliable.

Table 1: Adversarial defence performance of MI30 and VAR300. Error rate is the frequency of incorrectly classified samples. Pass rate is the frequency of samples passing the adversarial sample detection. Passed error rate is the frequency of incorrectly classified samples within the samples which pass the detection. Error rate with defence is the product of pass rate and passed error rate.

| MI/VAR | w/o defence | w/ defence | | |
|---|---|---|---|---|
| | error rate | pass rate | passed error rate | error rate |
| $\eta = 0$ | 0.0585/0.1250 | 0.9155/0.9720 | 0.0490/0.1325 | 0.0448/0.1288 |
| $\eta = 0.001$ | 0.1130/0.2061 | 0.9129/0.9758 | 0.0989/0.2086 | 0.0903/0.2033 |
| $\eta = 0.02$ | 0.9899/0.5577 | 0.0594/0.3140 | 0.9827/0.7865 | 0.0583/0.2470 |

## 5 DISCUSSION

In the experiments on synthetic data, we have two interesting discoveries. First, when we use GIN to learn the synthetic data, not only the informative variables can be successfully recovered as predicted by the theory, but also can the noise variables be learned. Actually, the features in the learned representation $\mathbf{w}$ is very close to a permutation of the true latent variables $(\mathbf{z}, \boldsymbol{\epsilon})$. This is demonstrated in Figure 9 by plotting the correlation matrix between $\mathbf{w}$ and $(\mathbf{z}, \boldsymbol{\epsilon})$. This result indicates that the identifiability theorem established by the current nonlinear ICA theory may be extended to a stronger version. In the identifiability theorem of linear ICA (Comon, 1994), the transforms allowed are restricted to linear ones while distributions of data are only required not to be Gaussian. Compared to the linear ICA theory, current nonlinear ICA theory requires a stronger assumption on latent distributions, which have to be in an exponential family, but has no constraints on the transforms. Note that in the synthetic data experiments, both the generating and the learning models are flows.

If we restrict the nonlinear transform to certain types of neural networks, we may be able to drop the exponential family assumption on their latent variables but still get identifiability.

More than the correct recovery of noise variables, we even find out that the variances of latent variables are approximately preserved, in particular when the variances of noise variables are much smaller ($10^{-4}$ for the following example) than those of the informative variables; see Figure 8. This may be caused by the fact that the greatest and least singular values of the Jacobian of flow-based models are both upper and lower bounded. Because of this phenomena, the variance criteria can find out the correct informative variables for low noise scenarios. Such a variance-preserving property is not stable when the variance of informative variables and noise have closer scales. Therefore, the mutual information criterion has more reliable performance in informative latent variables selection.

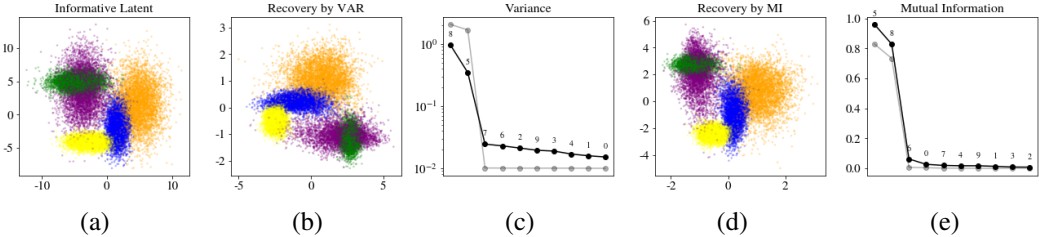

Figure 8: a) Ground truth informative latent dims. b) The top-2 features in $\mathbf{w}$ by the VAR criterion. c) Standard deviations of $z_i$'s and $w_i$'s. d) The top-2 features by the MI criterion. e) Mutual information of $z_i$'s and $w_i$'s.

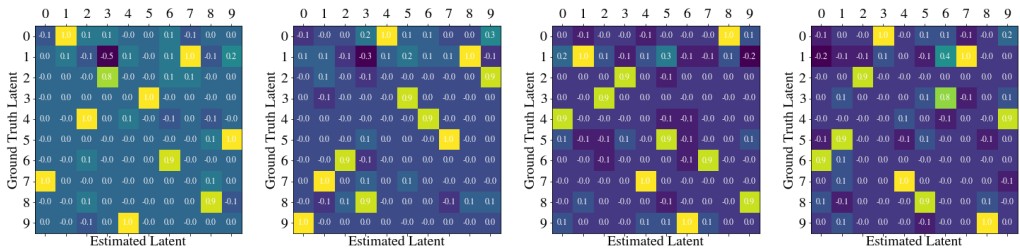

Figure 9: Four examples of correlation between ground truth latent (by rows), including both informative and noise dimensions, and estimated latent (by cols) of four different trained GIN models.

## 6    CONCLUSION

Based on our theoretical analysis and extensive experiments on synthetic and real data, we conclude that for any GIN type of models, the mutual information between representations and the auxiliary variable is a better criterion for informative latent variables selection than the previously proposed variance criterion. We also show that by selecting the correct features in the representation, the GIN model can perform better on the classification, outlier detection and adversarial defence tasks. The capability of GIN models perfectly recovering of all latent variables up to a permutation and a bounded scaling on synthetic data indicates the possibility of a stronger version of identifiability of nonlinear ICA using flow-based models. More importantly, the perspective that the auxiliary variable $u$ provides a context for the informative latent variables selection eliminates the boundary between informative latent variables and noise. It is thus possible that different auxiliary variables define different group of informative latent variables. This will lead to representation of data with a finer structure.

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

# A  NONLINEAR ICA

In this section, we provide the formal statements about the identifiability of Nonlinear ICA in Sorrenson et al. (2020) to give a context of discussions in the paper.

In the problem setting of Nonlinear ICA, the data $\mathbf{x} \in \mathbb{R}^d$ is assumed to be generated by a latent variable $\mathbf{z} \in \mathbb{R}^n$ through a nonlinear function $f$ with a noise variable $\epsilon$. There are two different formulations about the generation process, where both formulations would normally allow $n \leq d$. By assuming $n \leq d$, the framework is able to model data with intrinsic/informative support on a lower dimensional manifold.

- In Khemakhem et al. (2020)'s setting, the nonlinear function $f$ is assumed to be an injection from $\mathbb{R}^n$ to $\mathbb{R}^d$ and another noise variable $\epsilon \in \mathbb{R}^d$ is applied after $f$, i.e.,

$$\mathbf{x} = f(\mathbf{z}) + \epsilon. \tag{7}$$

- In Sorrenson et al. (2020)'s setting, the nonlinear function $f$ is assumed to be a bijection from $\mathbb{R}^d$ to $\mathbb{R}^d$ and another noise variable $\epsilon \in \mathbb{R}^{d-n}$ is concatenated to $\mathbf{z}$ before $f$, i.e.,

$$\mathbf{x} = f(\mathbf{z}, \epsilon). \tag{8}$$

In this work we only discuss the second setting.

Apart from $f$ being injective in the first setting and bijective in the second, normally we do not put more assumptions on the generating function $f$. As indicated by the name of ICA, we would require independence among entries in the latent variable $\mathbf{z}$. Although for linear ICA problems one usually assumes the mutual independence on entries in $\mathbf{z}$, one would usually introduce an auxiliary variable $u$ and assume the conditional independence $z_i|u \perp z_j|u$ for any $i \neq j$ due to obtain identifiability.

The identifiability of the latent variables is at the core of independent component analysis theory. It is worth noting that simple transforms, e.g., scaling, shifting and permutation, of the underlying true latent variables do not hurt the indepedence/conditional independence among them. As a result, the identifiability is actually considered under the equivalence relationship induced by these transforms.

Next we review the results from Sorrenson et al. (2020). We refer to the relationship between the ground truth latent variables and data as the generation process and that between the estimated latent variables and data as the estimation process.

**Notation 1.** *(Generation) Assume $n \leq d$ and we do not have $\epsilon$ for $n = d$.*

1. *Denote by $\mathbf{x} \in \mathbb{R}^d$ the observable data.*

2. *Denote by $u$ the auxiliary variable which is also observable. $u$ can be either a scalar or vector.*

3. *Denote by $\mathbf{z} \in \mathbb{R}^n := \mathcal{Z}_1 \times \cdots \times \mathcal{Z}_n$ the informative latent variable*

4. *Denote by $\epsilon \in \mathbb{R}^{d-n}$ the noise variable.*

5. *Denote by $f$ a bijection from $\mathbb{R}^d$ to $\mathbb{R}^d$.*

**Assumption 1.** *(Generation)*

1. *Assume the observable data $\mathbf{x}$ is generated as $\mathbf{x} = f(\mathbf{z}, \epsilon)$.*

2. *Assume the informative latent $\mathbf{z}$ conditioned on the auxiliary variable $u$ admits an exponential family distribution*

$$p(\mathbf{z}|u) = \prod_{i=1}^{n} \frac{Q_i(z_i)}{Z_i(u)} \exp\left[\sum_{j=1}^{k} T_{i,j}(z_i)\lambda_{i,j}(u)\right], \tag{9}$$

*where $T_{i,j}$'s are the sufficient statistics. $\lambda_{i,j}$'s are the coefficients and $Z_i$'s are the normalising constants. This assumption implies the conditional independence between entries in $\mathbf{z}$.*

3. *Assume the distribution of $\epsilon$ must not dependent on $u$. This assumption indicates that $\epsilon$ has zero mutual information with $u$ by definition.*

**Notation 2.** *(Estimation)*

1. *Denote by $\mathbf{w} \in \mathbb{R}^d$ the estimated latent variable.*

2. *Denote by $g_\theta : \mathbb{R}^d \to \mathbb{R}^d$ a bijection from a parametric family with parameter $\theta \in \Theta$. We may omit the parameter $\theta$ for notation simplicity.*

**Assumption 2.** *(Estimation)*

1. *Assume there exists a $\theta^\dagger \in \Theta$ such that $\mathbf{x} = g_{\theta^\dagger}(\mathbf{w})$.*

2. *Assume the estimated latent variable $\mathbf{w}$ conditioned on $u$ also admits an exponential family distribution*

$$p(\mathbf{w}|u) = \prod_{i=1}^{n} \frac{Q_i'(w_i)}{Z_i'(u)} \exp\left[ \sum_{j=1}^{k} T_{i,j}'(w_i)\lambda_{i,j}'(u) \right], \qquad (10)$$

*where $T_{i,j}'$'s are the sufficient statistics. $\lambda_{i,j}'$'s are the coefficients and $Z_i'$'s are the normalising constants.*

**Theorem 1.** *(Sorrenson et al., 2020) In addition to Assumption 2 and 1, we further assume:*

1. *The sufficient statistics $T_{i,j}(z)$'s are differentiable almost everywhere and their derivatives $\frac{dT_{i,j}(z)}{dz}$ are nonzero almost surely for all $z \in \mathcal{Z}_i$ and all $1 \le i \le n$ and $1 \le j \le k$.*

2. *There exist $nk + 1$ different values of $u$, i.e., $u_0, \ldots, u_{nk}$ such that the matrix*

$$L = [\boldsymbol{\lambda}(u_1) - \boldsymbol{\lambda}(u_0), \ldots, \boldsymbol{\lambda}(u_{nk}) - \boldsymbol{\lambda}(u_0)],$$
$$\text{where } \boldsymbol{\lambda}(u) = (\lambda_{1,1}(u), \lambda_{1,2}(u), \cdots, \lambda_{n,k}(u))^\top \qquad (11)$$

*is invertible.*

*Then the sufficient statistics of the generating latent space are related to those of the estimated latent space by the following equation:*

$$T(\mathbf{z}) = AT'(\mathbf{w}) + \mathbf{c}, \qquad (12)$$

*where $A$ is a constant, full row rank $nk \times dk$ matrix and $\mathbf{c}$ is a constant vector.*

From Theorem 1, we can see that the estimated latent variables are connected to the ground truth latent variables by the unique linear relationship. Without further knowledge about the matrix $A$, entries of $\mathbf{z}$ are still entangled in $\mathbf{w}$ in general. However, for two-parameter exponential family distributions, each row in the mixing matrix $A$ only has one non-zero element (Sorrenson et al., 2020). Here we recall the results for Gaussian distributions and refer interested readers to Sorrenson et al. (2020) for more results.

**Theorem 2.** *(Sorrenson et al., 2020) Assume both $p(\mathbf{z}|u)$ and $p(\mathbf{w}|u)$ are Gaussian distributions, then each row in $A$ only has one non-zero entry and each column has at most one non-zero entry. In other words, for each $z_i$ in $\mathbf{z} = [z_1, \ldots, z_n]$, there exist unique $w_j$, $a_i \ne 0$ and $b_i$, such that*

$$z_i = a_i w_j + b_i. \qquad (13)$$

From Theorem 2, we can see that for each $z_i$, there is only one $w_j$ estimating it up to an affine transform.

## B  GENERAL INCOMPRESSIBLE-FLOW NETWORK

A General Incompressible-flow Network (GIN) is a flow-based model integrating the features of both RealNVP (Dinh et al., 2017) and NICE (Dinh et al., 2015).

Flow based models, a.k.a. normalising flows, describe discrete dynamics for random variables and thus induce transports of probabilistic densities via change of variable. In the continuous time limit,

neural networks can also be used to model the time derivatives and thus define neural ODEs (Chen et al., 2018a). The absolute value of the determinant of the Jacobian matrix in the formula of change of variable usually requires highly complex computation. The design of flows calls for the balance between sufficient capacity of the model family for flexible modelling and sophisticated structure of the models for efficient computation. RealNVP approaches the problem by stacking coupling layers with form

$$\begin{aligned} \mathbf{y}_1 &= \mathbf{x}_1 \\ \mathbf{y}_2 &= \mathbf{x}_2 \odot \exp(s(\mathbf{x}_1)) + t(\mathbf{x}_1) \,, \end{aligned} \tag{14}$$

where the input $\mathbf{x}$ and output $\mathbf{y}$ are respectively divided into two parts $(\mathbf{x}_1, \mathbf{x}_2)$ and $(\mathbf{y}_1, \mathbf{y}_2)$. Due to the clever construction, evaluation of the Jacobian's determinant is reduced to the summation of $s(\mathbf{x}_1)$. The computation of the determinant can be completely eliminated by constructing $s(\cdot)$ to maintain a zero sum. In the coupling structure of NICE, there is no elementwise multiplication on $\mathbf{x}_2$, which de facto defines a zero sum $s(\cdot)$. The GIN paper proposed to output the last entry of $s(\cdot)$ as the negative sum of previous entries. As summing over huge amount of dimensions might cause numerical issues in extreme cases, we replace it with a mean subtraction mechanism

$$\hat{s}(\mathbf{x}_1) = s(\mathbf{x}_1) - \frac{1}{p} \sum_i s(\mathbf{x}_1)_i \,, \tag{15}$$

where $s(\mathbf{x}_1)_i$ is the $i$-th entry of the output vector of $s(\mathbf{x}_1)$ and $p$ is dimension of $s(\mathbf{x_1})$. A later release of the official GIN implementation also employs this mechanism independently. Note that by using zero-output-sum network, we could not only save the computation of the Jacobian's determinant but also preserve the volume of the object passing the flow.

## C    TRAINABLE STATISTICS VERSUS EMPIRICAL STATISTICS

As we mentioned in Section 3, there are two different objectives proposed by GIN: trainable statistics versus batch evaluations. To compare the performance of these two approaches, we also train the batch evaluation version of GIN on synthetic data for two different variance choices of $\boldsymbol{\epsilon}$: $\sigma_{\epsilon_i} = 1$ and $\sigma_{\epsilon_i} = 10^{-2}$. The results are shown in Figure 10 and Figure 11, and they are very similar to the results of trainable statistics in Figure 3 and Figure 8.

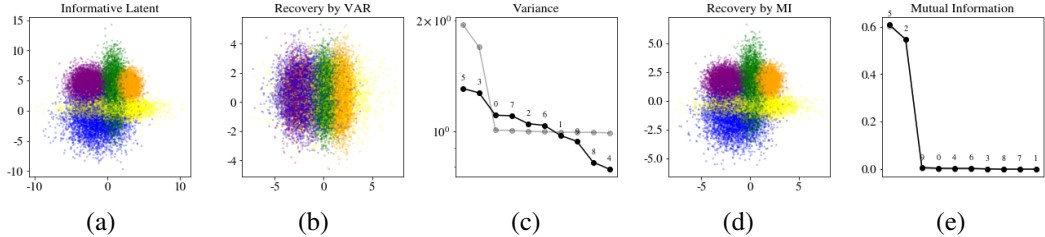

Figure 10: An example of GIN trained with empirical distribution parameters with high noise level. a) Ground truth informative latent dims. b) 2 dimensions with largest variance. c) Standard deviation decay. d) 2 dimensions with largest mutual information. e) Mutual information decay.

## D    EXPERIMENTS CONFIGURATIONS

Since the main scope of our paper is to compare the mutual information criterion with the variance criterion for informative latent variable identification and to show the benefit for downstream tasks, we adopt the same configurations of network structure and training methods with the original GIN paper. The networks are composed of serial coupling layers and dimension permutation layers. While the random permutation could allow different segmentation of the variable $\mathbf{x}$ and add additional flexibility on coupling layers, it will not affect the Jacobian determinant's evaluation. The permutations are applied on the length axis for synthetic data and the channel axis for the EMNIST data. For the EMNIST data, the coupling layers are mainly based on convolutional layers, while

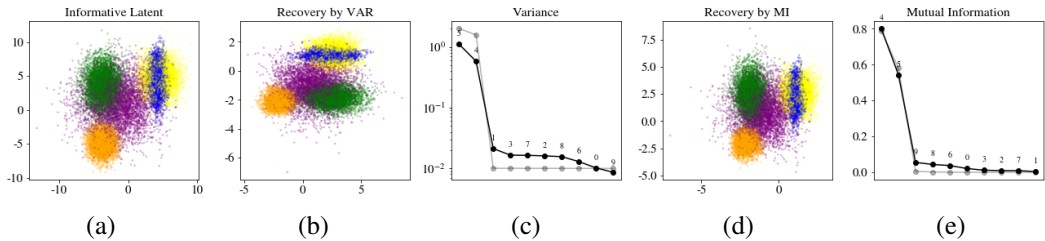

Figure 11: An example of GIN trained with empirical distribution parameters with low noise level. a) Ground truth informative latent dims. b) 2 dimensions with largest variance. c) Standard deviation decay. d) 2 dimensions with largest mutual information. e) Mutual information decay.

for the synthetic data, the coupling layers are based on fully connected layers. Besides, for EM-NIST data, the model also adopts invertible downsampling layers proposed in Jacobsen et al. (2018) and fully connected couplings after a flattening layer. More details can be found in Appendix D of Sorrenson et al. (2020).

Here we provide more information about the data used in our experiments.

For the synthetic data, the parameters of Gaussian mixtures are randomly generated from uniform distributions. As with the GIN paper, we use interval $[-5, 5]$ to sample the expectations and $[0.5, 3]$ to sample the standard deviation of $z_1, z_2$. For the one shown in Section 3, the statistics are

$$\begin{bmatrix} \mu_1 \\ \mu_2 \\ \mu_3 \\ \mu_4 \\ \mu_5 \end{bmatrix} = \begin{bmatrix} 2.9240 & 4.1936 \\ -3.8942 & -3.7325 \\ -4.1159 & 4.1676 \\ 4.3792 & 2.9258 \\ 4.4143 & -4.9151 \end{bmatrix} \quad \text{and} \quad \begin{bmatrix} \sigma_1 \\ \sigma_2 \\ \sigma_3 \\ \sigma_4 \\ \sigma_5 \end{bmatrix} = \begin{bmatrix} 1.7167 & 1.7170 \\ 2.5244 & 2.7930 \\ 0.8198 & 2.8419 \\ 2.3446 & 1.5909 \\ 0.9897 & 0.9298. \end{bmatrix} \quad (16)$$

And the distributions of OOD samples come from two Gaussian distributions with statistics

$$\hat{\mu}_1 = [-20, 20] \quad \hat{\sigma}_1 = [1.34, 1.35] \quad \text{and} \quad \hat{\mu}_2 = [20, 0] \quad \hat{\sigma}_2 = [1.21, 0.96]. \quad (17)$$

Figure 12 and 13 respectively show all the possible 2D projections of the ground truth latent variable $(\mathbf{z}, \boldsymbol{\epsilon})$ and observation $\mathbf{x}$ adopted in the results represented in Section 3. Figure 14 shows the estimated representation $\mathbf{w}$ by the trained GIN model shown in Section 3. While the correct informative latent dimensions are reconstructed within $\mathbf{w}$, the mutual information criterion can correctly identify them but the variance criterion cannot.

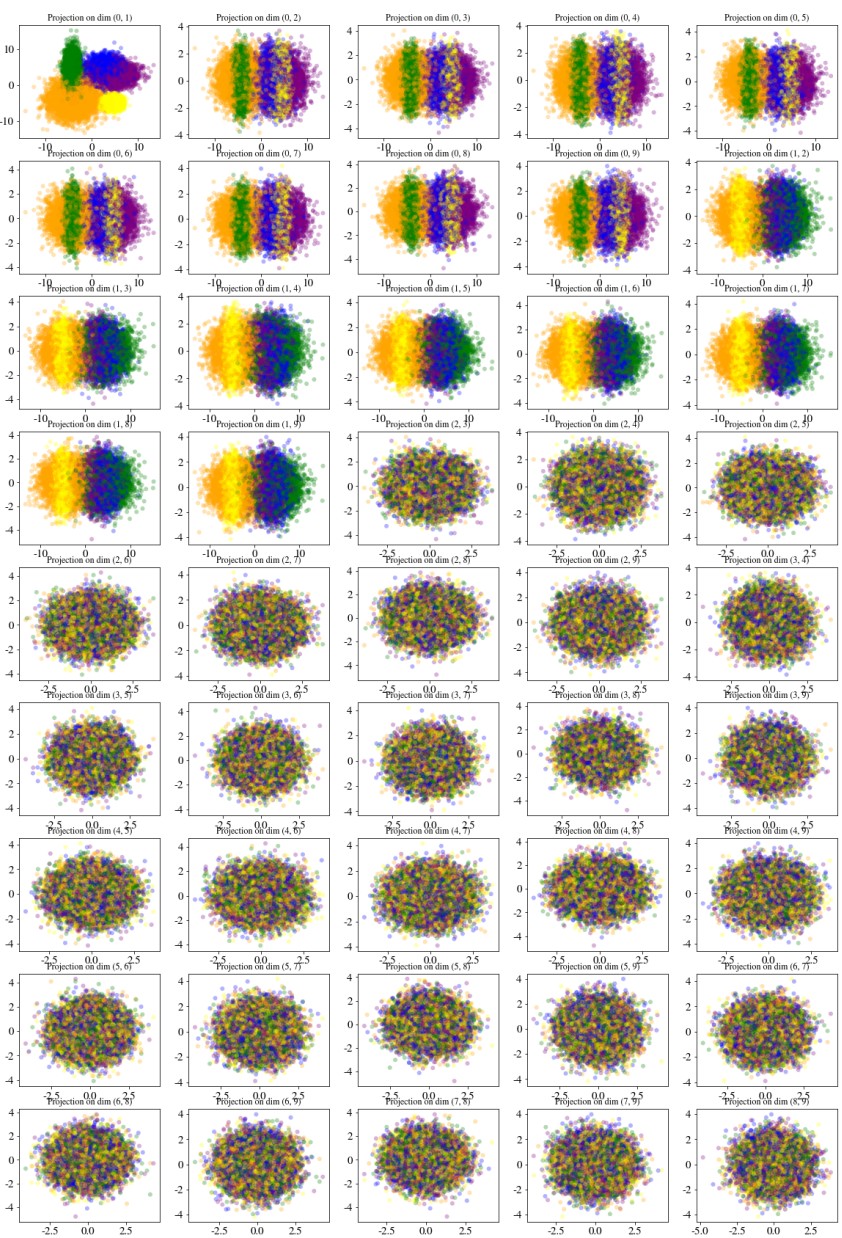

Figure 12: All 45 possible 2D-projections of the 10 dimensional latent $(\mathbf{z}, \epsilon)$ used in Section 3.

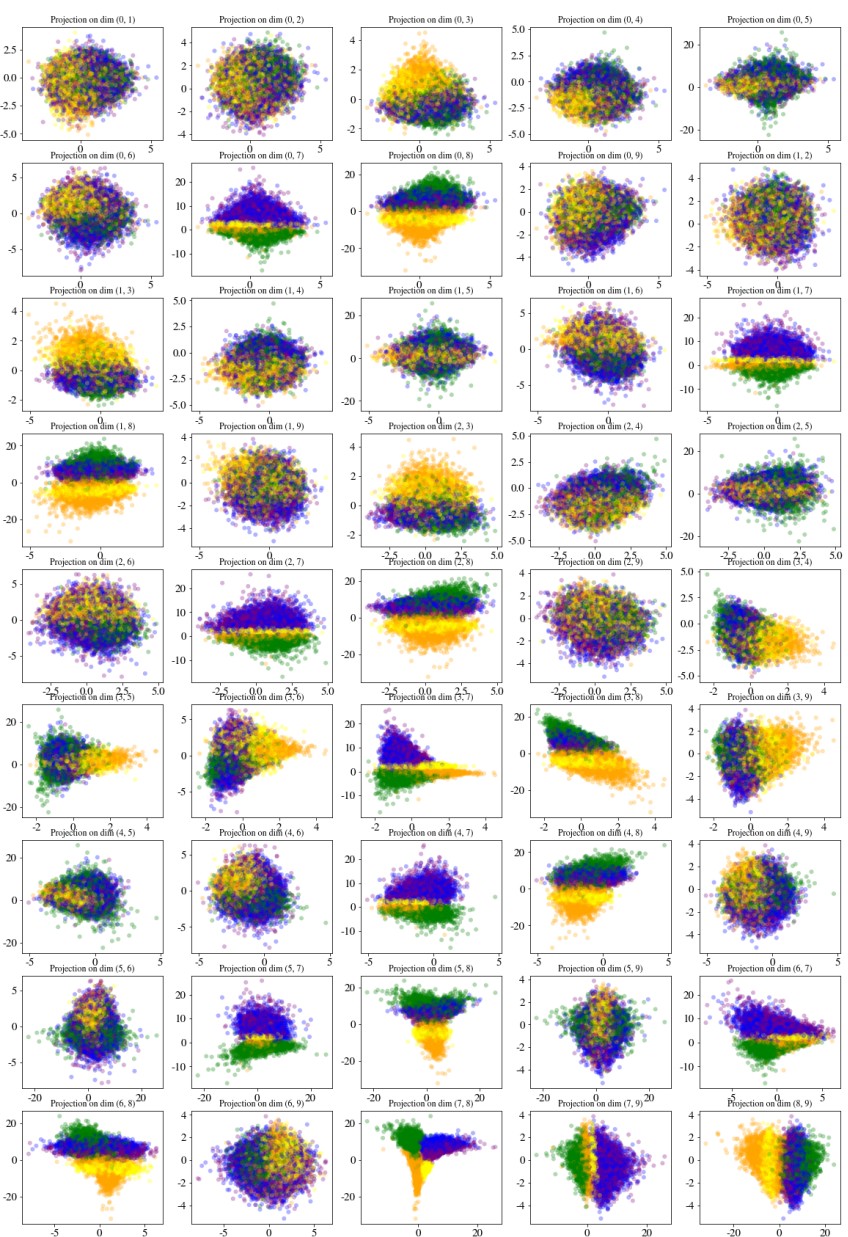

Figure 13: All 45 possible 2D-projections of the 10 dimensional observation $f(\mathbf{z}, \boldsymbol{\epsilon})$ used in Section 3.

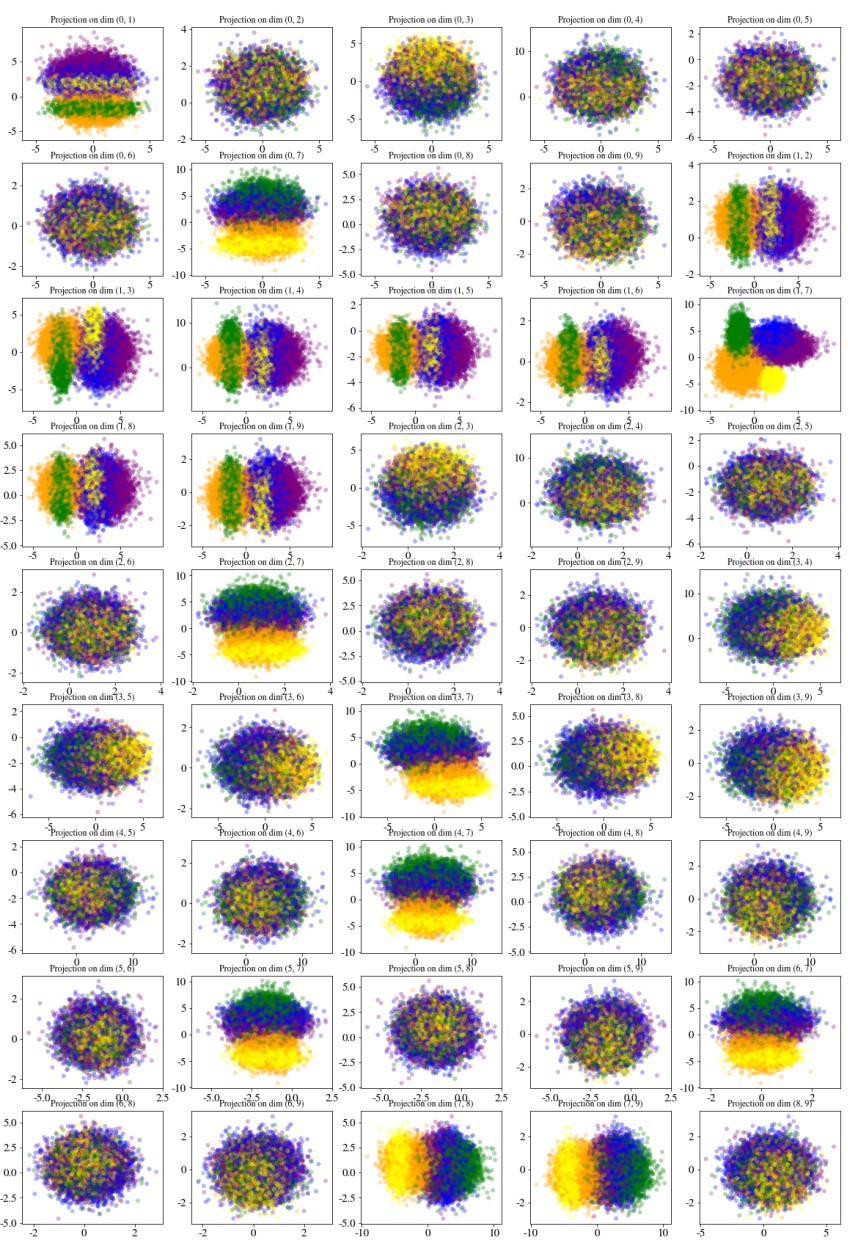

Figure 14: All 45 possible 2D-projections of the 10 dimensional estimated representation **w** used in Section 3.

