# OpenReview forum: "Identifying Informative Latent Variables Learned by GIN via Mutual Information"
_ICLR.cc/2021/Conference — Reject_

### Official Review · AnonReviewer4 · 2020-10-27

**Rating:** 5
**Confidence:** 3

**Review:**

## Summary

The authors propose an alternative method for finding informative latent variables in a model called General Incompressible-flow Networks (GIN). While previous work relied on the variance of the variables to assess informativeness, the authors argue that this is problematic when the scale of noise epsilon is large. They propose instead to use the mutual information between the variables and u instead.

The authors evaluate their alternative identification method on two tasks: a toy example of a mixture of gaussians and EMNIST. They show that when the variance of epsilon is large, then their method outperforms the variance based method (figure 3). However when the variance of epsilon is small, then VAR performs similarly to MI (Figure 8).

On EMNIST, it's clear that MI is able to identify the most important variables and outperforms the best VAR setting, however when more variables are included the MI method deteriorates quicker than VAR.

## Review

In general I find the paper difficult to follow. Many paragraphs have an unclear structure and sentences are not linked. There are also many unsubstantiated, or unclear/imprecise claims. The derivations and equations seem correct and the figures are well made and understandable. Especially figures 1 and 2 are informative and clear.

I think the paper would be a lot stronger if it was positioned as "This is a failure case of using the VAR method with GIN, we propose a rigorous solution". Currently the authors make statements such as "Though Sorrenson et al. successfully establish the identifiability, their interpretation on the meaning of this result and the method they propose are incorrect", "it is unplausible to use the variances of the learned representation" and "while the original VAR criterion is not competent at all" - despite clearly showing that the VAR method works well in a number of situations. I think it's worth being more nuanced about alternative methods and explicitly showing when the other method does not behave as described/expected (which you also do!).

Could the authors comment on how the MI was computed and in particular the computational trade offs between using MI and VAR to select variables? I'm also curious why the MI method degrades so quickly in figure 6, how did you set up the classification task?

How did you "apply defence" in the adversarial defence experiment? You mention that VAR300 is only fooled by 56% of the cases without defence, but this cannot be compared because its "clean accuracy" is lower, could you expand on that? It seems to me that higher accuracy is always better? It seems in contradiction with the claim at the end of the paragraph that "all results indicate that ... features selected by MI criterion are more reliable."

In general, a lot of the experiments seemed to be based on Sorrenson et al, but are not properly described in the paper. I would like to see at least a basic description.

In summary, I think the authors have identified an interesting improved that is worth publishing. However I think that the paper in its current form is not ready: it can benefit from significant rewriting for clarity (both in structure, as well as spelling) and further analysis of the behaviour in figure 6.

## Notes:

"Incompressive" --> incompressible
"in nonlinear independent analysis theory" - missing "component"?
W_{d+1:n} should be W_{n+1:d} (below equation 4)?
"unplausible" -> "Implausible"?
"interpretation on" -> "interpretation of"

There's a number of other, non-existent, words in the paper that will be caught by a spellcheck.

## After updating paper

After reading the updates, I think the authors have considerably improved their paper and I have increased my score.

While the core contribution, using MI over VAR, is clear. The evaluation is not strong enough, I still don't understand why adversarial defense is a reasonable way to evaluate the different metrics. Further it's now clear that the author's use the sklearn implementation of MI estimation, which depends on several entropy estimators which have high variance in practice. While the authors comment that it's "negligible" I am not convinced that this is actually easy or reliable.

In summary I think the paper is not ready for publication.

---

> ### Author Response · Authors · 2020-11-18
> **Thanks for the comments**
>
> We thank the reviewer for the comments and we would like to take this opportunity to clarify some confusions.
>
> #### **1. Reply to '...worth being more nuanced ...'**
> The authors find the previous work interesting and inspiring both theoretically and practically.
> Our main concern is about the variance based criterion to identify the informative latent, which is not supported by theory in all scenarios.
> It is an very interesting phenomenon that VAR works in low noise synthetic experiments we conducted.
> Nevertheless, the inconsistent results caused purely by the noise level and the contradiction with current identification theory indicate that VAR might not be a suitable criterion to identify the informative latent.
> We appreciate the advice on writing and will be more cautious in preparing the revision.
>
> #### **2. Reply to '..how the MI was computed...'**
> Our current algorithm is to compute and sort the mutual information between individual entries in $\mathbf{w}$ and $u$.
> Then we truncate the sorted space of $\mathbf{w}$ according to the mutual information decay curve.
> There are two schemes training the GIN framework, one is setting the distribution parameters trainable and the other one is evaluating the distribution parameters empirically on the training batches.
> While using VAR criterion for the latter scheme involves extra computation, one can directly use the trained parameter from the first scheme.
> Besides, evaluating variances is normally more efficient than evaluating mutual information on the same data batch.
> Nevertheless, since we only need to evaluate the mutual information once after training, the extra computation introduced by MI criterion would be negligible compared to the training time.
>
> #### **3. Reply to '... set up .. classification...'**
> The density based classification mechanism is the same for both synthetic and EMNIST data.
> As explained in Section 3, a sample $\mathbf{x}$ is classified into the category $u$ which can maximise the posterior probability $p_S(u|\mathbf{x})$.
> And the set of informative latent $S$ is selected respectively by VAR and MI.
> Let's say MI thinks in $\mathbf{w}=[w_1, w_2]$, $w_1$ is the informative variable, then $S=\{w_1\}$ and $p_S(u|\mathbf{x})$ is evaluated by $\{p(w_1|u_i), p(u_i)\}_{i\in I}$.
>
> In practice, one only need to select a truncation level and use it for classification.
> As a result, the quicker deterioration of larger truncation levels (possibly by including noise dimensions) for MI does not hurt its top performance for classification purpose.
> Moreover, as the low mutual information entries are very likely encoding no additional information to the previous ones and mainly encode noise, we should not include them for classification in principle.
> We think the most problematic entries have intermediate level mutual information (which do not encode new information and should not be included) and low level variance.
> And MI include them earlier than VAR, which leads to the sharp and early performance drop.
> When they include the same and all entries, they perform the same and worst.
>
> #### **4. Reply to 'how... apply defence'**
> Similar to the idea of using density models for classification, we design adversarial defence by identifying samples with low density on all categories as adversarial samples, as explained in Section 4.
> Following the above example, for a sample $\mathbf{x}$ with estimated latent $\mathbf{w}=[w_1, w_2]$, if $p(w_1|u_i)$ for any $i\in I$ is lower than certain thresholds, MI will refuse to classify it and achieve the defence.
> And we use the density value of the points one standard deviation away from the mean as thresholds as mentioned in Section 4.
>
> Our point in the adversarial attack experiments is to show that MI30 can admit a better adversarial defence mechanism than VAR300 with the density models they build by selecting different sets of estimated latents as informative latents.
> As a result, we state that we cannot prefer VAR300 simply because the error rate without defence is lower when $\eta=0.02$.
> The clean accuracy ($\eta=0$) of MI30 is higher than that of VAR300 is due to that MI admits a better density model.
> Since the decision boundary is defined by a density model, the sensitivity to adversarial attack is also defined by the density model.
> Although we do not have a rigorous answer, we conjecture MI30 degenerates more when $\eta=0.02$ is related to the decision boundary.
> We infer that MI30 decision boundary separates the latents with a narrower margin which admits higher clean accuracy and higher adversarial sensitivity.
> Moreover, thanks to the better density model, we could identify the adversarial samples and refuse to classify them.

---

> > ### Comment · AnonReviewer4 · 2020-11-20
> > **Reply**
> >
> > Thank you for your extensive and clear comment.
> >
> > I think the paper will be much stronger if you incorporate your answers to 1-4 in the text and I will update my score accordingly upon reading the revised paper.
> >
> > My main outstanding concern is the size of the contribution. It would be very interesting if there's more to gain from using MI over VAR aside from these evaluations. I understand that this might be out of the scope of the discussion period.

---

### Official Review · AnonReviewer1 · 2020-10-28

**Rating:** 6
**Confidence:** 4

**Review:**

This paper considers the previously established Generative Incompressible Flow (GIN) model to perform disentangled representation learning and argues that the original method of identifying explanatory latent variables via their variance magnitude is flawed. Identification is necessary because flow models require the same number of latent variables as the data dimensionality, thus many latent variables are expected to be uninformative, due to the general low-dimensional data manifold hypothesis. This paper proposes the mutual information between a latent variable and an auxiliary variable (needed because otherwise, the disentanglement objective is unsolvable in general) as an identification criterion and shows that this outperforms the variance criterion, especially in cases where the noise magnitudes are of the same order as the magnitudes of the explanatory variables.

I think this paper makes solid arguments for why the original variance criterion is flawed under certain conditions and why the mutual information criterion is better and I lean towards accepting.

Here are a couple of comments, weak points, and questions:
- Labeled axes would make the plots more readable without having to re-read the caption multiple times
- At some point, you claim that the importance of any latent variable can be dependent on which auxiliary variable is used. The claim seems to be that the same data could have a set of latent variables that would have high MI with u1, and another set that would have high MI with u2. I see that this is feasible (and even probable), but you never show this, not even in synthetic data. In fact, in the synthetic data, your 8 noise variables are very idealistic. I would therefore either take out the claims or include experiments that show these effects, both in synthetic and real data.
- In Figure 6, the performance decreases for MI as you include more variables. You discuss this in the text, but it is not satisfactory to me. Could you formulate more clearly what you mean here? Is the degradation an artifact of not being able to estimate these variables well? Or is your core assumption of a low-data manifold violated? Or could it be that the MI is a better, but still not the correct criterion?
- How do you estimate the MI, in detail?

---

> ### Author Response · Authors · 2020-11-18
> **Much appreciated for the constructive comments**
>
> We would like to thank the reviewer for very constructive comments.
> We hope to first partially address the reviewer's concerns and open more discussions, while working on a writing improved manuscript.
>
> #### **1. Reply to 'the importance of any latent variable ...'**
> To make sure we correctly understand the reviewer's comment, we would like to expand on our statements in the paper with an example.
> Assume the data $\mathbf{x}$ is generated by a nonlinear function $f$ with an informative vector $\mathbf{z}_1$, another informative vector $\mathbf{z}_2$ and noise $\boldsymbol\epsilon$ such that $x=f(\mathbf{z}_1, \mathbf{z}_2, \boldsymbol\epsilon)$.
> Assume that we have one auxiliary variable $u_1$ and another auxiliary variable $u_2$, and $u_i$ only has non-zero mutual information with $\mathbf{z}_i$.
>
> Then we can ask several interesting questions: 1) can we reconstruct both $\mathbf{z}_1$ and $\mathbf{z}_2$ with only $u_1$ or $u_2$?, 2) can we reconstruct $\mathbf{z}_1$ and $\mathbf{z}_2$ with the Cartesian product auxiliary $u=(u_1, u_2)$? etc...
>
> We are interested in these questions but feel they can be out of the main scope of this paper.
> As a result, we use 'may have' in the paper to point out the possibility instead of making a certain assertion.
>
> #### **2. Reply to 'In figure 6, ...'**
> Since in practice, one only need to select a certain truncation level and use it for classification, the quicker deterioration of larger truncation levels (possibly by including noise dimensions) for MI does not hurt its performance for classification purpose.
> Moreover, the non-informative estimated latents in $\mathbf{w}$ should not be used for classification in principle and our experimental results also validate this point.
>
> Nevertheless, we are interested in this phenomenon and would like to share some of our thoughts on it.
> As there is no ground truth informative latent of EMNIST and the theoretical requirements of Nonlinear ICA are not satisfied by the dataset, we couldn't directly justify if the variables are or even can be estimated well.
> For the low dimension manifold assumption for EMNIST, we tend to accept it as there are also other works arguing the similar point, e.g., Facco et al.
> And we also think it may be possible that current individual MI evaluation algorithm is not perfect.
>
> Among many possible reasons to the quicker deterioration phenomenon, the one presented in our current submission is from the perspective of using density models for classification.
> We can see that when MI and VAR include the same all entries, they achieve the same and worst performance.
> We conjecture that the most problematic entries may have intermediate level of mutual information but low level variance.
> As a result, they are included into $S$ much earlier by MI than by VAR and cause sharp drop of performance in Figure 6 for MI.
> Assume $w_{150}$ is one of the entries causing sharp drop.
> We think it might be possible that $p(w_{150}|u_i)$'s, $i\in I$, have lots of overlaps to each other.
> As a result, a point in $u_i$'s cluster might have a higher density value in on $p(w_{150}|u_j)$ and this mismatch contributes a lot to the calculation to $p_S(u|\mathbf{x})$.
>
> #### **3. Reply to 'how do you estimate MI ...'**
> Our algorithm is to compute and sort the mutual information between individual entries in $\mathbf{w}$ and $u$.
> For the synthetic data, we can use a large number of samples to evaluate the mutual information, while for the EMNIST experiments we are using a batch with size 100.
> Then we truncate the sorted entries of according to the mutual information decay curve.
>
>
> E. Facco, M. d’Errico, A. Rodriguez, and A. Laio. Estimating the intrinsic dimension of datasets by
> a minimal neighborhood information. Scientific reports, 7(1):1–8, 2017.

---

### Official Review · AnonReviewer3 · 2020-10-28
**A good improvement of informative latent variables selection in general incompressible-flow networks, but incremental for publication**

**Rating:** 5
**Confidence:** 3

**Review:**

This paper considers the problem of disentangled representation learning with an existing normalizing-flow-based approach, called general incompressible-flow networks (GIN). In the original approach, informative latent variables were separated from noise by considering the variances of learned latent variables and selecting ones with high variances. The current paper shows empirically that this approach can fail even when the underlying data generating process satisfies all assumptions needed for identifiability. The reasoning is simple: Latent variables can have different scales and their variances may not be indicative of informativeness. Instead, the current paper proposes to use the Shannon mutual information between the auxiliary observed variable (conditioned on which the true underlying factors of variation are conditionally independent and belong to the exponential family) and latent variables to separate informative latent variables from noise. With experiments on synthetic data and on EMNIST, the paper successfully shows that the proposed mutual-information-based approach works better than the original variance-based approach (in terms of recovering the ground truth latent factors, downstream classification accuracy, and out-of-distribution detection).

The paper achieves a good improvement in informative latent variables selection in GIN, but is not a significant contribution for ICLR publication. To improve the paper, the authors may consider proving that the mutual-information-based approach recovers the ground truth factors in the setting of equation (1) or (2). Additionally, the proposed approach of selecting informative latent variables can be considered more broadly in the context of other disentangled representation learning approaches.

# Update
Thanks for the rebuttal. I have read the changes the authors did. The updated manuscript is more readable and more self-contained now. I am still of the opinion that the presented method should be put in a broader context (i.e., considering it in broader settings and/or for other methods) or be better analyzed theoretically. This way it will be much more useful for the community. For this reason, I keep the score the same.

---

> ### Author Response · Authors · 2020-11-17
> **Thank you for the comments and we are curious about the broader context of disentangled representation learning**
>
> We appreciate the positive and constructive comments from the reviewer and thank the reviewer for getting our main message.
>
> #### **1. Reply to 'consider proving ...'**
> Our current algorithm is to compute and sort the mutual information between individual entries in $\mathbf{w}$ and $u$.
> For the synthetic data, we can use a large number of samples to evaluate the mutual information, while for the EMNIST experiments we are using a batch with size 100.
> Then we truncate the sorted entries of  according to the mutual information decay curve.
>
> With the assumptions of identifiability holding true, one can further claim that the non-informative latent entries are independent on $u$.
> As a result, they will admit zero mutual information with $u$ which validate our algorithm.
>
> #### **2. Reply to 'more broadly in the context of ... disentangled representation learning...'**
> As our work now is mainly motivated in the specific context of Nonlinear ICA with flow based models, the authors are not very sure how it can be used in a broader disentangled representation learning context.
> Therefore, we would love to ask the reviewer to kindly elaborate more.

---

### Official Review · AnonReviewer2 · 2020-10-29
**Ambiguous writing obscures the contribution**

**Rating:** 4
**Confidence:** 3

**Review:**

This paper proposes a method to select informative latent variables for representation learning. The idea is to find variables that maximize mutual information with respect to observed auxiliary variables. The paper points out that in Sorrenson et al. the latent variable selection criteria does not lead to good guarantees, and empirically the new selection criteria has superior performance.

Pro:

The paper points out the problem that since the scale of the latent variable cannot be recovered, the scale cannot be used as a criteria for variable selection. This seems true and an important point to note.

The proposed alternative variable selection criteria seem reasonable, but the algorithm and its theoretical properties need to be rigorously stated.

Con:


The writing is quite difficult to follow. One big issue is that this paper is not self contained, i.e. it inherits the notation in prior work (especially Sorrenson et al) without explaining what the symbols mean. Many notations are also confusing. For example in Eq.(5) what does w(z,eps) mean? I can kind of get you are trying to represent the dependence of w on z and eps, but this expression does not type check.

How is the mutual information criteria implemented in practice? I’m assuming you want to find a subset of variables that maximize mutual information w.r.t. u, but this is a problem known to be difficult (i.e. a naive solution has to enumerate every subset of variables). I inferred that you are finding **individual** variables that maximize mutual information with respect to u; However there is a big gap between Eq.(5) and the claim that you can maximize mutual information of each individual variable w.r.t. u. How is this justified, or did I misunderstand? I think you need to explicitly say exactly what algorithm you are using, and what property it should have.

The experimental settings are also not clearly explained. As a few confusing sentences (among many others), "each latent variable is labeled by auxiliary variable" or "the signal provided by the informative latent variables may not be able to outstand". I don't think I can understand what is going on exactly here with so much ambiguity.

---

> ### Author Response · Authors · 2020-11-17
> **Thanks for the comments and we are improving the clarity**
>
> We sincerely thank the reviewer for getting our main message despite the writing being ambiguous.
> While we are working on a more self-contained and clearer manuscript based on the reviewers' feedback, we would like to take the opportunity to address part of the reviewer's concerns.
>
> #### 1. **Reply to 'How is the mutual information ... implemented in practice'**
> Our current algorithm is to compute and sort the mutual information between individual entries in $\mathbf{w}$ and $u$.
> For the synthetic data, we can use a large number of samples to evaluate the mutual information, while for the EMNIST experiments we are using a batch with size 100.
> Then we truncate the sorted entries of $\mathbf{w}$ according to the mutual information decay curve.
>
> Now we motivate our MI criterion from more straightforward and clear way in Section 2.
> And under this motivation, it can be guaranteed that we can correctly discover the informative latent variables as long as the identifiability theory's assumptions hold.

---

### Official Review · AnonReviewer5 · 2020-11-06
**Reservations arise about the exposition and content of the paper**

**Rating:** 6
**Confidence:** 2

**Review:**

This paper builds on top of the paper “Disentanglement by nonlinear ICA with General Incompressible-Flow Networks (GIN)” (Sorrenson, 2020) and argues that that paper’s method of identifying informative latent variables was wrong and instead suggests that informative latent variables can be identified by thresholding their mutual information with the auxiliary variable of the conditional generative model.

Overall, I score this paper as a reject. Without judging the content as reason, the exposition of the paper is overly difficult to understand. It also makes very specific references to Sorrenson (2020) and Khemakhem (2020) and their proofs without providing context (Section 2). Similarly, it introduces nonlinear ICA without defining the model well (Section 1). This reviewer had to read the mentioned papers side-by-side to make sense of the paper under review.

### Comments about the content

GIN/Nonlinear ICA wants to recover the true generative latent variables $\mathbf{z}$ for data $\mathbf{x}$ given the conditioning variable $\mathbf{u}$, which acts as the cause for the true latent. In particular, it would be important to mention that both data $\mathbf{x}$ and auxiliary $\mathbf{u}$ are observed. An example is given in Sorrenson (2020) for (E)MNIST: $\mathbf{u}$ is the digit to draw, $\mathbf{z}$ are the parameters that guide the drawing, $\mathbf{x}$ is the output. The mentioned papers show that the true latent variables can be recovered up to affine transformations, and additional latent variables are noise.

This paper sets out to examine the question on how to separate informative latent variables from latent variables that encode noise (in the recovered latent) and claims that Sorrenson (2020) chooses the wrong metric for thresholding (they use the std deviation of the different latent variables). This reviewer could not find any particular mentions of this question in Sorrenson (2020), except for implicit usage in the experiment section.

However, this paper claims that mutual information is a better metric and Sorrenson (2020) made a mistake in their proof. This reviewer did not retrace the proofs. Moreover, this reviewer could not understand how eq. (5) in this paper, which uses the data processing quality with the joint of the latent, motivates using the mutual information of the individual latent variables to identify informative ones.

Moreover, intuitively, the MI with the auxiliary variable will tell us about the dependence of the two, which will identify latent variables that encode local information for the conditioning variable (eg digit) and which change a lot in their distribution depending on the conditioning variable (using the language from Sorrenson (2020)).

This leads to the question about “global” parameters, which do not change in distribution. For example, if MNIST was not grayscale but had digits with different colors (equally likely), the color could be a global parameter, which would be uncorrelated with the conditioning variable and thus have zero mutual information. This would mean that this paper would treat it as a noise variable, which it is not.

As such, this reviewer would ask the authors for clarification.

### Additional comments

1. The figures lack axis labels.
2. FGSM is a rather weak adversarial attack.
3. Section 5 and Figure 9 mentions that the learnt representation $\mathbf{w}$ is very close to a permutation of the real latent variables. This is explained in Sorrenson (2020), Section 3.2: “when both the generating and estimated latent spaces follow a Gaussian distribution, the latent space variables are recovered up to a trivial translation and scaling.” This is the case here: there is no rotation, which leads to a permutation in the correlation matrix.

---
This reviewer wants to thank the authors for their detailed reply and for updating the paper to make it more self-contained. It reads much better now and is much clearer. The review score has consequently been updated from 4 to 6. Stronger adversarial experiments would be encouraged. Could the authors also include a definition of the VAR criterion (even though it is trivial, just to avoid any ambiguities) and maybe include a paragraph for the future applications of this? For this reviewer, it is not entirely clear yet what the value of this finding is: is it going to help with downstream tasks for ICA? Is this another argument in favour of preferring Mutual Information as a metric over variances in general?

---

> ### Author Response · Authors · 2020-11-17
> **While improving for a more self-contained manuscript, we would like to first clarify part of the concerns raised by the reviewer.**
>
> We sincerely thank the reviewer for the effort reading this submission.
> While preparing a more self-contained manuscript, we would like to first clarify part of the concerns raised by the reviewer.
>
> #### **1. Reply to 'This paper sets out to examine the question ...'**
> Since the dimension of estimated latent $\mathbf{w}$ by normalising flows is equal to or higher than the dimension of the informative latent $\mathbf{z}$, it is a natural and necessary question to ask how to identify a subset of entries in $\mathbf{w}$ as the informative estimated latent. Sorrenson (2020) did nice work on obtaining $\mathbf{w}$ and revealing that $\mathbf{z}$ is hiding in $\mathbf{w}$, while our work studys how to correctly identify $\mathbf{z}$ from $\mathbf{w}$.
>
> #### **2. Reply to '... how eq. (5) in tha paper...'**
> Now we explain our motivation in a more straightforward and clear way.
>
> #### **3. Reply to '... global parameters...'**
> The Assumption (ii) in Theoreom 1 of Appendix A in Sorrenson (2020)  requires the matrix $\mathbf{L}$ to be invertible.
> One can verify that if any entry in $\mathbf{z}$ has distribution parameters invariant to the conditioning variable $u$, the matrix $\mathbf{L}$ will not be invertible.
> As a result, the 'global parameters' mentioned by the reviewer are regarded as noise by the very setting of current Nonlinear ICA theory, not only by our proposed criterion.
>
> #### **4. Reply to ' FGSM ... weak ... attack'**
> The main point of the paper is not to show that we can achieve state-of-the-art adversarial robustness by using MI criterion compared to other methods in the adversarial literature.
> Instead, we aim at showing compared to VAR criterion, MI criterion can admit a better adversarial defence mechanism under the same attack type and level.
>
> #### **5. Reply to 'Section 5 and Figure 9 .....'**
> From Reply 2, one could see that Sorrenson (2020) Section 3.2 only guarantees the relationship between $\mathbf{z}$ and entries in $\mathbf{w}$ that 'linearly' estimate $\mathbf{z}$, while in our paper, Section 5 and Figure 9 pointed out such phenomenon between the whole $\mathbf{w}$ and the tuple $(\mathbf{z}, \boldsymbol\epsilon)$.
>
> Even if one neglects the discussion in Reply 2, one must assume that the Gaussian noise entries are part of $\mathbf{z}$, in which case $n=10$.
> While the Assumption (ii) in Theoreom 1 of Appendix A in Sorrenson (2020) requires at least $nk+1=10*2+1=21$ different values of $u$, we only have 5 in our experiments.

---

### Author Response · Authors · 2020-11-24
**Revision summary**

The authors appreciate the constructive comments from all reviewers and we would like to summarise the revision of our paper here.

- Axis labels in the figures are added.
- Appendix A with more detailed nonlinear ICA review is added to make the paper more self-contained.
- Section 2 is restructured to make the MI algorithm and its motivation more clear.
- More details about how we conduct classification are added in Section 3
- More discussion on the classification accuracy decay and Section 4

---

### Decision · Program_Chairs · 2021-01-07
**Final Decision**

**Decision:**

Reject

**Comment:**

The paper proposes a method to identify informative latent variables by thresholding based on the conditional generative model. While the exposition of the paper has substantially improved during the discussion period, some major concerns remain after the discussion among the reviewers. In particular, the problem considered in the paper has a very limited scope. Moreover, the evaluation of the methods needs to be improved. The paper could benefit from discussing how it situates in the broader context.